# Differential Expression of Genes Involved in Metabolism and Immune Response in Diffuse and Intestinal Gastric Cancers, a Pilot Ptudy

**DOI:** 10.3390/biomedicines10020240

**Published:** 2022-01-23

**Authors:** Martine Perrot-Applanat, Cynthia Pimpie, Sophie Vacher, Ivan Bieche, Marc Pocard, Véronique Baud

**Affiliations:** 1INSERM U1275, CAP Paris-Tech, Université de Paris, Lariboisiere Hospital, F-75010 Paris, France; cynthia.crocheray@inserm.fr (C.P.); marc.pocard@inserm.fr (M.P.); 2Pharmacogenomics Unit-Institut Curie, Department of Genetics, Université de Paris, F-75005 Paris, France; sophie.vacher@curie.fr (S.V.); ivan.bieche@curie.fr (I.B.); 3Hepato-Biliary-Pancreatic Gastrointestinal Surgery and Liver Transplantation, AP-HP, Pitié Salpêtrière Hospital, F-75013 Paris, France; 4NF-kappaB, Différenciation et Cancer, Université de Paris, F-75006 Paris, France

**Keywords:** immune checkpoint, tryptophan metabolism, gastric cancers (GCs), diffuse GC, intestinal subtype GC, and aryl hydrocarbon receptor (AhR)

## Abstract

Gastric cancer (GC) is one of the major causes of cancer-related mortality worldwide. The vast majority of GC cases are adenocarcinomas including intestinal and diffuse GC. The incidence of diffuse GCs, often associated with poor overall survival, has constantly increased in USA and Europe The molecular basis of diffuse GC aggressivity remains unclear. Using mRNA from diffuse and intestinal GC tumor samples of a Western cohort, this study reports the expression level of the immunomodulatory aryl-hydrocarbon receptor (AhR), and genes involved in immune suppression (*PD1*, *PD-L1*, *PD-L2*) and the early steps of tryptophan metabolism (*IDO1*, *IDO2*, *TDO2*). Strongly increased expression of *IDO1* (*p* < 0.001) and *PD1* (*p <* 0.003) was observed in the intestinal sub-type. The highest expression of *IDO1* and *PDL1* correlated with early clinical stage and absence of lymphatic invasion (×25 *p* = 0.004, ×3 *p =* 0.04, respectively). Our results suggest that kynurenine, produced by tryptophan catabolism, and AhR activation play a central role in creating an immunosuppressive environment. Correspondingly, as compared to intestinal GCs, expression levels of *IDO1-TDO2* and *PD-L1* were less prominent in diffuse GCs which also had less infiltration of immune cells, suggesting an inactive immune response in the advanced diffuse GC. Confirmation of these patterns of gene expression will require a larger cohort of early and advanced stages of diffuse GC samples.

## 1. Introduction

Gastric cancer (GC) is a major health problem and one of the major causes of cancer-related mortality worldwide [1] with a high incidence in Asia [2]. However, GC is a highly heterogeneous disease in terms of classification, clinical presentation and epidemiology. The vast majority of GCs are adenocarcinomas, which can be further histologically classified in intestinal-, diffuse- and mixed types according to the Lauren classification [3]. The majority of intestinal subtype GC arises from chronic gastritis and is associated with infectious agents including *Helicobacter pylori* and Epstein-Barr virus (EBV). The incidence of intestinal subtype subtype of GC has been declined over the past 50 years, possibly as the result of the decreasing prevalence of *Helicobacter pylori* [4]. In contrast, the incidence of the diffuse subtype of GC has constantly increased among Western populations (0.1 to 1.4/year for 100,000 habitants between 1973 and 2000 in USA) [5,6]. The diffuse sub-population is unrelated to *Helicobacter pylori* and develops from morphological normal gastric mucosa without atrophic gastritis. Most patients with diffuse GC, especially so for the signet-ring cell carcinoma (SRCC), tend to present with an advanced-stage disease with lymphovascular invasion, frequent metastasis and poor overall survival [7,8,9,10]. Given the poor prognosis of diffuse cancer, advances in cancer biology and molecular profiling are needed to elucidate the molecular basis of growth and metastasis in advanced GCs.

Diffuse GCs were often associated with germ line mutation in *CDH1* or *RhoA* amplification [11,12,13,14] and aggressive behavior [8,9,15,16]. Several studies have reported the activation of oncogenic signalling pathways in diffuse GC, such as hedgehog-EMT, Wnt/β catenin signalling, along with the expression of PI3K/Akt responsive genes [2,9,17]. Moreover, a TGFβ-associated supermodule of stroma-related genes associated with late stage diffuse type morphology has been reported [18]. Our group has also identified new genes including mesenchymal markers (*IGF1*, *FGF7*, *TGFβ* and *ZEB2*, *CXCR4*) whose expression is associated with aggressive phenotype of diffuse GC [14]. In this study we have explored two other signalling pathways, namely immunosuppressive genes and metabolic reprogramming in both intestinal and diffuse GCs.

Most of our knowledge of the immune context in cancer derives from studies on melanoma, lung, breast, colon, and prostate cancers, as well as from animal models. Cancer cells may exhibit immune inhibition to promote tumor progression and distant metastasis. One key mechanism is the Programmed Cell Death 1 (PD1)/PD1 ligand (PDL1) pathway [19,20]. PD1 is activated by its ligands PDL1 and PDL2 to suppress antigen-stimulated lymphocyte proliferation, migration and cytokine production, resulting in attenuation of effector T cells function and immunological tolerance [20]. Recent studies from Asian groups have reported the clinical implication of variations of the levels of immunosuppressive proteins such as PD-L1 and PD1 in patients with GCs [21,22,23]. A molecular characterization of clinical response to PD-1 inhibition in metastatic gastric cancer indicated a favourable response in EBV and MSI GC [22,24]. The last decade has also witnessed the emergence of novel therapeutic targets and combination strategies to address advanced HER2-positive GC [25]. In contrast little is known about the expression and role of immune cells in GC of the diffuse subtype GC.

Metabolic reprogramming is a hallmark of cancer and considered to be critical to support accelerated proliferation, progression and metastasis. Imbalance in tryptophan (Trp) metabolism is found in several cancers, and kynurenine (kyn) is recognized as a critical microenvironment factor that contributes to immune depression [26]. IDO1 (2,3-dioxygenase) and TDO2 are two intracellular enzymes that mediate the first and the rate-limiting step of tryptophan catabolism in the kynurenine pathway [27,28,29]. Activation of the IDO1 pathway induces the blocking of differentiation, affects the functional anergy of effector T cells and promotes the de novo differentiation of Treg. Moreover, the correlated expression of *IDO1* and collagen genes synergistically enhances tumor cell migration and invasion in vivo and in vitro [30]. While *TDO2* expression by tumor cells themselves has been reported in several carcinomas, the role of TDO2 remains unclear.

Aryl hydrocarbon receptor (AhR) is a ligand-activated member of the PAS family of basic-helix-loop helix transcription factor. Through binding to exogenous and endogenous ligands, AhR has been involved in important cellular and pathological processes, such as control of proliferation, migration, angiogenesis and tumorigenesis [31,32,33]. An important role has emerged for AhR as a receptor for the endogenous ligand kynurenine [27] and for environmental ligands, and as a modulator of cancer immunity [29,34,35,36,37,38,39]. The role of AhR in immune escape program remains to be investigated in GCs.

In this pilot study, the primary objective was to document the expression of genes involved in the immunosuppressive PD1/PDL1 pathway (*PD1*, *PDL1*, and *PDL2*), tryptophan metabolism (*IDO1* and *TDO2*), along with the immunoregulator AhR in a cohort of GCs patients [14], comparing aggressive/diffuse and intestinal subtypes. We investigated the dynamic expression of these genes during GC progression. We also analysed a possible link between the immune checkpoint *PD1/ PDL1*, *IDO1* and *AhR* expression in gastric cancers.

## 2. Material and Methods

### 2.1. Patients and Tissue Samples

The cohort of 29 GC patients has been previously described [14]. In addition to gender, age, tumor size and depth of tumor invasion, lymphatic invasion, TNM status and smoking were introduced. The malignancy of infiltrating carcinomas was scored according to the TNM staging system (Stage I to IV), first according to AJCC7, revised from IGCA and AJCC8. This TNM staging includes a T score in the primary tumor (T1-T4), N score (lymph node metastasis) and M (metastasis) (see Table 1).

### 2.2. Total RNA Preparation and Real-Time RT-PCR

The conditions for total RNA extraction, complementary cDNA synthesis and qRT-PCR conditions were as previouly described [14]. We used real-time quantitative PCR to analyse the expression of selected genes in the gastric tumors samples as compared to the non-tumoral samples. The theoretical and practical aspects of real-time quantitative PCR have been described in detail elsewhere [14]. using ABI Prism 7900 Sequence Detection System (Applied Biosystem; Thermo Fisher Scientific, Inc., Waltham, MA, USA). Reverse transcription PCR was conducted with the high capacity cDNA reverse transcriptase kit (Applied Biosystem; Thermo Fisher Scientific, Inc, Waltam, MA, USA). We quantified transcripts of the *TBP* gene (Genbank accession NM 003194) encoding the TATA box-binding protein (a component of the DNA-binding protein complex TFIID) as an endogenous housekeeping gene, and normalized each sample to the *TBP* content, as previously described.

Primers for genes were selected using the Oligo 6.0 computer program (National Biosciences, Plymouth, MN, USA). We searched the dbEST and nr databases to confirm the absence of single nucleotide polymorphisms in the primer sequences and the total gene specificity of the nucleotide sequences chosen as primers. The nucleotide sequences of the primers used to amplify target genes are available on request. Each sample was normalized on the basis of its *TBP* content. Results, expressed as N-fold differences in target gene expressions relative to the *TBP* gene (and termed “*Ntarget*”), were determined as *Ntarget* = 2^ΔCtsample^, where the ΔCt value of the sample was determined by subtracting the average Ct value of the specific target gene from the average Ct value of the *TBP* gene. *Ntarget* values of the samples were subsequently normalized so that the median of *Ntarget* values for normal gastric tissues (*n* = 11) was 1. Preliminary analysis of gene expression have compared basal levels in normal samples in the same patients as their tumors (either diffuse- or intestinal- GC subtypes). We did not observe changes for most of the genes described in the study (ratio for the median levels ranging from 0.8 to 1.2). Moreover, to increase the reliability of the method of detection, gene expression was measured simultaneously. For each gene expression, normalized RNA values of 3 (or more) were considered to represent gene overexpression in tumor samples, and values 0.33 (or less) represented gene underexpression.

### 2.3. Statistical Analysis

For each gene, differences of expression between tumors versus normal tissues (fold change) were analyzed as previously described [14,29]. The relative expression of genes was characterized by the median and the range. Differences in the number of samples that over- (>3-fold) or under- (<3-fold) expressed were analyzed using the Chi2-square test. The relationships between expressions of genes in gastric cancer were determined using non parametric Spearman’s rank correlation test. Relationships between expression levels and clinical parameters were analyzed using non parametric Kruskal-Wallis (or Mann-Whitney) and Chi-square tests, as indicated in each Table. Statistical analyses were performed using Prism 5.03 software (GraphPad, San Diego, CA, USA). Differences were considered significant at confidence levels greater than 95% (*p* < 0.05).

### 2.4. Immunocytochemistry

Immunohistochemical labeling was performed on paraffin sections (4 mm) as previously described [29]. Immunohistochemical analysis for AhR (santaCruz) was performed using Ventana Autostainer (Roche Diagnostics, Indianapolis, IN, USA). AhR immunostaining was analyzed blindly by two specialists including a certified pathologist.

## 3. Results

### 3.1. Patient’s Characteristics

The clinicopathologic characteristics of the study population are shown in Table 1. The distribution of gastric tumor subtypes was as follows: diffuse (*n* = 13, 45%) and intestinal (*n* = 16, 55%) subtype GCs, according to the Lauren classification. Patients with diffuse adenocarcinoma are younger (*p* = 0.0004, Table 1), and harbor tumors with more aggressive characteristics, such as more lymphatic invasion (Table 1
*p* = 0.001), accompanied by massive stromal fibrosis [14] and metastasis than patients with the intestinal GC sub-type (Table 1). Vascular and neural invasion were not different (Table 1). In addition, when comparing the TNM stage, diffuse GC was present at TNM stages II, III and IV (38%, 31% and 31%, respectively), while intestinal subtype was more likely at stages I, II and III (26%, 44% and 25%, respectively).

### 3.2. Expression of PD-L1, PD-L2 and PD1 in Gastric Cancers

Significant higher expression of *PD-L1* was observed in intestinal subtype vs. peri-tumoral samples (Table 2), along with overexpression (>3) in 30% of the cases (Appendix A). Significant higher *PD-L1* expression in the intestinal subtype occurred in the absence of lymphatic invasion (×3, *p* = 0.04) and in the early stages (TNM I-II, ×2, *p* = 0.03) as compared to non-tumoral tissue (Table 3). In contrast to the intestinal subtype, no significant modulation of *PD-L1* expression was observed in diffuse GC vs. peri-tumoral samples (no overexpression and no change with clinical parameters) (Table 2, Table 3 and Appendix A).

The expression of *PD-L2*, encoding another ligand for PD1, was increased in both intestinal and diffuse GC subtypes (×1.67, *p =* 0.014 and ×1.21 *p =* 0.036, respectively) (Table 2), but with no cases of high expression (>3) vs. their peri-tumoral samples (Appendix A). In the intestinal subtype, a higher expression of *PD-L2* correlated with the absence of vascular invasion (*p =* 0.03) and smoking (*p <* 0.001) (Table 3). *PD-L2* was independent on clinical parameters in diffuse GCs (Table 3).

*PD1* encodes the receptor for PD-L1 and PD-L2. Expression of *PD1* significantly increased in gastric tumors vs. their peri-tumoral counterpart (×1.63, *p =* 0.001), both in intestinal and diffuse subtypes (×1.71, *p = 0.003* and ×1.53 *p = 0.009,* respectively) (Table 2). Overexpression *of PD1* (>3) was observed in 19% of intestinal and 8% of diffuse GC (Appendix A). Interestingly, in intestinal GCs, a higher *PD1* expression was observed in the absence of lymphatic invasion (×2.2, *p =* 0.05), absence of perineural invasion (×3.5, *p =* 0.02) and lower TNM stages (I-II) (×2.2, *p =* 0.037) (Table 3) that are clinical parameters of early stage GCs, i.e., less aggressive GCs. In contrast to the intestinal GC, *PD1* expression was independent of clinical parameters in diffuse GCs (Table 3). Of note, in all GC subtypes, the mRNA expression levels of *PD1* did not differ according to sex, age, or vascular invasion (Table 3).

Overall, in the intestinal GCs, expression *of PD-L1* together with *PD1* gradually decreased from the early stage to the advanced stage (lymphatic invasion and/or TNM III-IV), along with a decrease in *PD-L2* expression with vascular invasion. In contrast, no correlation of *PD1*, *PD-L1* or *PD-L2* was observed with clinical parameters in diffuse GCs.

### 3.3. Expression of IDO1, IDO2 and TDO2 in Gastric Cancers

We further analyzed the expression of *IDO1, IDO2* and *TDO2*, three genes that encode enzymes involved in the early steps of tryptophan metabolism leading to kynurenine, an endogenous AhR ligand [27]. The tryptophan (TRP) to kynurenine (KYN) metabolic pathway is now firmly established as a key regulator of innate and adaptative immunity [26].

As shown in Table 2, the expression of *IDO1* was significantly increased in all gastric tumors vs. their peritumoral counterpart (×2.2, *p* < 0.0001), with *IDO1* expression being significantly higher in both the intestinal (×3, *p =* 0.0006) and diffuse (×1.96, *p =* 0.002) GC subtypes as compared to non- tumoral gastric tissues (Table 2). Heterogeneous expression of *IDO1* was observed in the intestinal subtype with a strong overexpression (>3) in 50% of cases, as compared to diffuse GC (23% of cases) (Appendix A). In all GC tumors, higher *IDO1* expression was significantly observed in less advanced stages, corresponding to absence of lymphatic invasion (×6.3, *p =* 0.005) and lower TNM stages (I and II, ×3.2, *p =* 0.036) (Table 4). Moreover, the higher expression of *IDO1* in intestinal subtype corresponded to the absence of lymphatic invasion (×25.4, *p =* 0.004) and to lower TNM stage (×6.4, *p =* 0.02) (Table 4). In contrast to intestinal GC, *IDO1* expression appeared independent of clinical parameters in our cohort of diffuse GCs (Table 4).

A significant increase of *TDO2* expression was observed in all GC tumors (×5.4, *p <* 0.0001), both in intestinal and diffuse GC subtypes (×7.4, *p* < 0.0001 and ×3.3, *p =* 0.0002, respectively), as compared to non-tumoral gastric tissues (Table 2). Strong overexpression (>3) of *TDO2* was preferentially observed in intestinal subtype (80% of cases, especially in the absence of lymphatic invasion (Table 4 and Appendix A). In the diffuse GCs, the highest expression of *TDO2* (with 50% of overexpression) was observed in males (*p =* 0.035) and in the absence of metastasis (*p =* 0.034) (Table 4). *IDO2* was expressed at very low basal level in non-tumoral gastric tissues. 

### 3.4. Correlation of Expression between PD-L1 and IDO1 in Gastric Cancers

As an exploratory analysis, we conducted non-parametric Spearman rank correlation tests to assess the associations of *PD1/PD-L1* and *IDO1* expression in gastric cancers (see Materials and Methods). The expression levels of *PD-L1* and *IDO1* were correlated in all GCs (r = 0.65, *p =* 0.0001) (Appendix A), both in the intestinal (r = 0.68, *p =* 0.004) and diffuse (r = 0.63, *p <* 0.05) GCs (Appendix A). The correlation between *PD-L1* and *TDO2* was lower in all GCs (r = 0.50, *p =* 0.006) (Appendix A), in the intestinal subtype (r = 0.32, *p* = 0.23) or in diffuse GCs (r = 0.48, *p >* 0.05) (Appendix A). Moreover, strong *PD1* and *PD-L1* correlated in intestinal subtype, while *PD1* and *PD-L2* correlated in diffuse subtype (Appendix A).

### 3.5. High AhR Expression in Gastric Cancers

Both IDO1 and TDO2 are a source of kynurenine, an activated ligand of AhR [26,27]. Therefore, since we observed a higher expression of *IDO1* and *TDO2* in gastric cancers as compared to non-tumoral tissue, we further analyzed *AhR* expression and protein localization in gastric tumors (see Materials and Methods). Increased *AhR* expression was found in GCs, both in intestinal and diffuse subtypes (×1.6, *p* = 0.003, and ×2.1, *p* = 0.001, respectively) (Table 2), with few cases of overexpression (8–12%, Appendix A). Moreover, *AhR* expression was independent of clinical parameters in all tumors, either intestinal or diffuse GCs (Table 5). AhR was present in tumor epithelial and stromal cells (Figure 1), including fibroblasts, endothelial and immune cells (such as lymphocytes).

## 4. Discussion

Intestinal and diffuse GCs are two gastric cancers with different aggressivity and prognosis. Intestinal GCs are more commonly diagnosed in aged patients, and are strongly associated with gastric mucosal atrophy and intestinal metaplasia, both of which are induced by chronic *Helicobacter pylori* infection. The diffuse subtype which represents a small minority of gastric cancer, is genomically stable and associated with decreased expression of *CDH1* (E-cadherin), higher expression of *RhoA* and prominent mesenchymal features, thus resulting in tumor aggressiveness [11,12,13,14,15,17]. Diffuse GCs are associated with frequent metastasis in lymph nodes and the peritoneum, contributing to their poor prognosis. Most anti-cancer therapies have failed to substantially improve prognosis of GC patients. The molecular driver of anti-tumor immunity in GCs is still poorly understood, posing a major obstacle for selection of GC patients for immunotherapy trials. Although the expression of immunosuppressive markers including *PD-L1* has been reported for various tumor types [21,40,41], it remains unclear for GCs and their associated subtypes in the Western population. In this retrospective analysis of tumor samples from 29 intestinal and diffuse GCs patients, from patients who underwent primary surgery at Lariboisiere Hospital (Paris, France)*,* we report for the first time, the expression levels of AhR, and of several genes involved in immune gene signature (*PD1, PD-L1* and *PD-L2*) and in tryptophan metabolism (*IDO1/IDO2, TDO2*). Association (or absence of association) with various clinical parameters are also described.

Escape from anti-tumor immunity is a second generation cancer hallmark. One key mechanism in the heterogeneous immune response is the PD1-PD-L1/PD-L2 axis. Immune checkpoints have been identified on both immune cells and tumor cells. PD1 is expressed on the surface of activated CD4+ and CD8+ T lymphocytes, B lymphocytes, but also on NK and dendritic cells [20]. PD-L1 is expressed on the surface of cancer cells and in cells of the tumor environment (T and B cells, macrophages and dendritic cells), while PD-L2 expression is more restricted to activated dendritic cells and macrophages [42]. The interaction of PD1 with ligands PDL1 and PDL2 provides an immune effector T cells and immune tolerance [20]. To date, data regarding the expression of *PD1* /*PDL1* in GC have been mostly evaluated in Asian populations [21,22,23]; studies in Caucasians are urgently needed.

In the present study, we present gene signatures associated with the PD-1-PD-L1/PD-L2 axis in GCs, including intestinal and diffuse subtypes. Selected gene expression differed between subtypes of GCs. A higher expression of *PD-L1* along with *PD-1* expression was observed in patients with intestinal GC, interestingly at a less advanced time (without lymphatic invasion and at the lower TNM). *PD-L1* expression was correlated with *PD1* expression. While the relevance of the PD1-PDL1 pathway in cancer has been extensively studied, the relevance of PD-L2 has received less attention. As compared to patients with a history of smoking, nonsmoking patients exhibit higher expression of *PD-L2* in intestinal GC, an observation that has been recently reported for *PD-L1* in lung cancer [43]. Therapy targeting both PD1 ligands may provide clinical benefit in these patients [44]. PD-L1 positive immune cells revealed by immunocytochemistry and infiltration of immune (CD3^+^ /or CD8^+^) cells correlated with survival outcome in Asian GC patients that include EBV-positive and MSI GCs [21,22,23,45]. More prevalent *PD-L1* expression and better patient outcome was also observed in Western patients with EBV and MSI [46]. EBV and MSI were the most infiltrated GCs, harboring 30–50% T cells and 20% macrophages, while intestinal GC contained fewer T cells and more macrophages [9,47,48].

In contrast to the intestinal subtype, lower expression of *PD-L1* was observed in diffuse GC (*p* = 0.02). Diffuse GC was the least infiltrated subtype GC. In particular, CD8^+^ T cells (CD8^+^ TILs) and circulating NK cells and Tregs were significantly lower in diffuse advanced gastric cancer compared to the intestinal type (*p* = 0.009) [9,49]. Low infiltration of immune cells, associated with low expression of *PD-L1* in diffuse gastric cancers, suggests an underdeveloped immune resistance. Taken into account that the rate of diffuse GCs has strongly increased among Western populations [6], this immune signature may suggest that individual evaluation of PD1, PD-L1 or PD-L2 would be ineffective.

Mice overexpressing an active AhR exhibit enhanced stomach cancer [50], suggesting a role of AhR in carcinogenesis [51,52,53]. We observed significant increased *AhR* expression in GCs as compared to non-cancerous tissues, independently of intestinal or diffuse GC subtypes and clinical parameters. Nuclear AhR present in GC tumor and immune cells, fibroblasts and endothelial cells (Figure 1), suggests an activation of AhR, as previously reported in breast cancer [29]. In the past several years, AhR has been established as a critical ligand-dependent transcription factor involved in the regulation of the immune system and inflammatory response [29,54,55]. A large variety of exogenous (present in food and environment) and endogenous AhR ligands provide a complex scenario of the impact of AhR on tumorigenesis and immune homeostasis [33,56]. The effect of AhR ligands on differentiation of Th17 and Treg occurs through different mechanisms [57]. Considering AhR ligands, tryptophan catabolism plays a central role in creating an immunosuppressive environment [28,34,36,58,59,60].

The levels of expression of *IDO1*, *IDO2* and *TDO2* involved in the early steps of tryptophan metabolism leading to the AhR ligand kynurenine, were analysed [27,28,29]. Interestingly, *IDO1* levels, but not *IDO2* levels (not expressed), were significantly elevated in high *AhR* expressing gastric tumors. Increased *IDO1* expression was mainly observed in the intestinal GC subtype, with an overexpression (>3) in 50% of the cases, along with a less advanced stage characterized by an absence of lymphatic invasion (*p* = 0.004) and lower TNM (*p* = 0.02)]. Higher *IDO1* expression in early stage intestinal GC subtype (Table 4) is consistent with a previous study [21]. *IDO1* can be overexpressed in tumor cells and dendritic cells, macrophages and endothelial cells [60]. IDO1 has been shown to be upregulated in an inflammatory microenvironment (e.g., in the presence of IFNγ, the most potent *IDO* inducer, LPS and pathogens) [61,62]. In other cancers, IDO1 expression may be constitutive with IDO-producing tumors cells surrounded by a lower number of lymphocytes. Our results suggest that the lower *IDO1* expression observed in diffuse GC as compared to the intestinal subtype (*p* = 0.009) may be related to the significantly lower circulating NK cells and Tregs described in (advanced) diffuse GCs [9]. IDO enzyme activity may lead to a local amino-acid starvation response. T cells and NK cells are very sensitive to tryptophan deprivation and downstream metabolites from IDO activity (kynurenine) in their microenvironment [60,63]. Two related populations of CD4+T cells (Th17/CD4+/CD25+ and regulatory T (Treg) cells), with opposing functions during immune responses, shifted from TH17-dominant (through inflammation) to Treg-dominant (TGFβ) according to GC progression [64,65].

We report for the first time significant *TDO2* up-regulation in GCs, both in intestinal (*p* = 0.014) and diffuse (*p* = 0.05) subtypes. The non-redundant role of IDO1 and TDO2 still remain unclear [66]. TDO2 was found in several cancers including lung, bladder, breast and ovarian carcinoma [29,67]. TDO2 is expressed in tumor cells that produce sufficient intracellular kynurenine concentrations to chronically activate the AhR, and by pro-inflammatory cytokines such as IL-6 in triple-negative breast cancer [67]. Notably, in triple negative breast cancer, the TDO2-AhR signaling axis promotes metastasis and resistance to anoikis [67]. The correlations between the expression of *TDO2* and other genes (e.g., *IDO1, TGFβ*, and *MMP9*) observed in diffuse GC, but not intestinal GC (Sup Table 3 and Table 4), suggests a role of tumor and immune cells that merits further consideration using a larger sample of patients.

Our study has some limitations. First, the small number of tumor samples (30) could be a limiting factor and could induce a bias between intestinal and diffuse subtypes. However, in a previous study with the same Western cohort [14], we observed comparable decrease or increases of the gene expression, notably *CDH1*, *CXCR4* and *TGFβ*, which are involved in epithelial mesenchymal transition and chemotaxis) in diffuse gastric cancers, as now well described. Second, we have not differentiated the subpopulation of diffusely infiltrating type of GC associated with extensive fibrosis (linitis or SRCC) [5,6] as compared to non linitis diffuse GC.

In conclusion, the results reported here document for the first time the time-dependent expression of PD-L1 and the IDO1-TDO2-kyn-AhR signaling pathway in diffuse and intestinal GCs. Our previous results in the same cohort of patients have suggested that mesenchymal features are more prominent in diffuse GC, resulting in tumor aggressiveness and fibrosis [14]. The present study suggests an inactive immune response in the advanced diffuse GC in patients undergoing surgery (no adaptive immune resistance). Further studies in a larger series of gastric tumor samples, especially with different clinical characteristics (-early diffuse subpopulation, and -SRCC known as an increased risk of developing peritoneal metastasis [68]), would offer opportunity to confirm genes of interest in aggressive GC. The nature of the signal (exogenous or endogenous) that drives AhR activation in diffuse GC has yet to be understood. The role of the surrounding tumor environment in gastric cancer is particularly important in tumor progression and metastasis. Actually, the microenvironment is a limitation factor for any drug penetration. The more we will be able to understand the different mechanisms implicated, the more we will be able to develop new therapeutic solutions. We can hope that in 5 years the classification of gastric cancer will be changed and the clinician could offer a specific microenvironment dedicated drug for the patient.

## Figures and Tables

**Figure 1 biomedicines-10-00240-f001:**
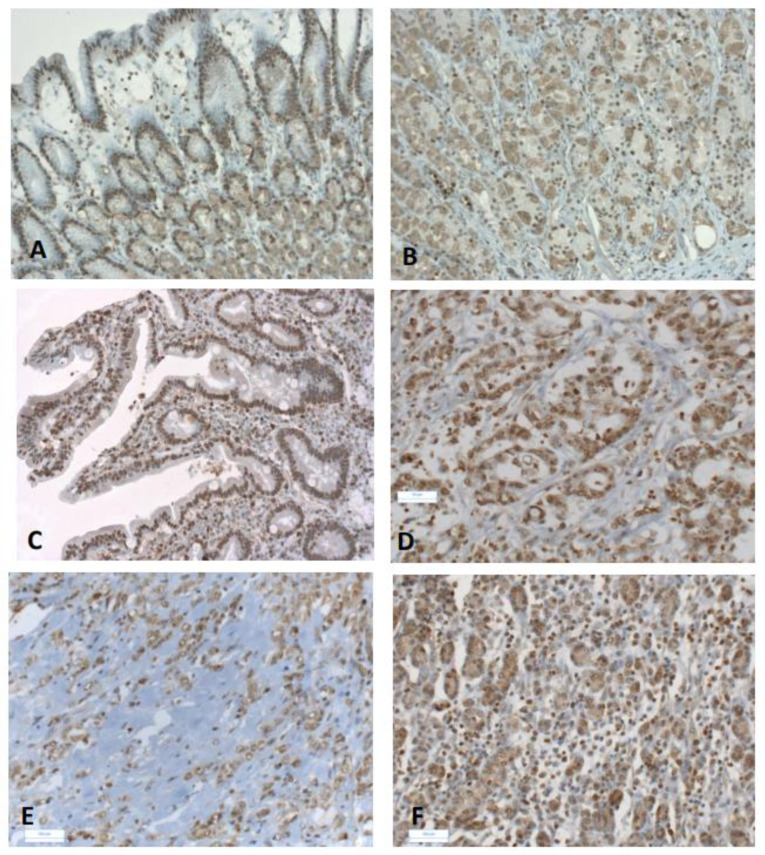
Immunohistochemical staining of AhR in gastric cancers. Representative immunostaining of AhR in non-tumoral gastric mucosa (**A**,**B**) and in gastric cancers (**C**–**F**). Weak cytoplasmic and nuclear expression of AhR were observed in epithelial cells (**A**,**B**). Intestinal subtype GC with metaplasia (TNM 2a) (**C**); strong nuclear AhR staining in epithelial and stromal cells (**C**). Moderately differentiated intestinal subtype (TNM2a) showing nuclear AhR staining in tubular glands and stroma (**D**). Advanced diffuse GC (TNM4) (**E**): the intensity of AhR immunostaining was lower in the scattered cells of single ring cell component (SRCC). Early diffuse GC (TNM2a) (**F**): AhR immunostaining in epithelial and stromal cells. Original magnification ×10 (**A**,**C**,**E**); ×20 (**B**,**D**,**F**).

**Table 1 biomedicines-10-00240-t001:** Clinicopathological characteristics of gastric carcinoma patients: poorly cohesive adenocarcinoma and intestinal-subtype adenocarcinomas. Median (range) of gene mRNA expression levels; *p* value (^a^ Chi2. ^b^ Mann Whitney). Significant *p* value < 0.05 (in bold), NS, not significant. Comparative basal levels of genes in normal tissue (×1) are as follow: *PD1* (19), *PDL1* (53), *PDL2* (86), *IDO1* (70), *TDO2* (18) and *IDO2* (1).

	Total GC (*n* = 29)	Poorly CohesiveGC (*n* = 13) (45%)	Intestinal-Subtype GC (*n* = 16) (*n* = 55%)	*p*-Value
**Gender, *n* (%)**				
**male**	13/29	6/13 (46%)	7/16 (43%)	0.90 (NS) ^a^
**female**	16/29	7/13 (54%)	9/16 (56%)	
**Age (years, median)**	63 +/−17	57(27–71)	75(59–82)	**0.0004 ^b^**
**Tumor size** **(mm), *n***				
**<50**	10/27	4/11 (36%)	6/16 (37%)	0.10 (NS) ^b^
**>=50**	17/27	7/11 (64%)	10/16 (63%)	0.95 (NS) ^a^
**Depth of tumor invasion**				
**T1-T2**	6/29	2/13 (15%)	4/16 (33%)	0.5 (NS) ^a^
**T3-T4**	23/29	11/13 (85%)	12/16 (67%)	
**Lymphatic invasion, *n* (%)**				
**negative**	11/28	1/13 (7%)	10/15 (67%)	**0.0014 ^a^**
**positive**	17/28	12/13 (92%)	5/15 (33%)	
**Vascular invasion, *n* + (%)**				
**negative**	9/29	3/13 (23%)	6/16 (38%)	0.67 (NS) ^a^
**positive**	20/29	10/13 (77%)	10/16 (62%)	
**Neural invasion, *n* (%)**				
**negative**	23/29	2/13 (15%)	4/16 (25%)	0.66 (NS) ^a^
**positive**	6/29	11/13 (68%)	12/16 (75%)	
**Metastasis (M), *n* (%)**				
**negative**	24/29	9/13 (69%)	15/16 (94%)	0.14 (NS) ^a^
**positive**	5/29	4/13 (31%)	1/16 (6%)	
**TNM status**				
**I-II**	16/29	5/13 (38.5%)	11/16 (69%)	0.10 (NS) ^a^
**III-IV**	13/29	8/13 (61.5%)	5/16 (31%)	
**Smoking**				
**negative**	12/22	4/12	8/12	0.77 (NS) ^a^
**positive**	10/22	3/10	7/10	

**Table 2 biomedicines-10-00240-t002:** Statistical analysis of mRNA expression of genes involved in immunity and tryptophan metabolism in gastric cancers. Median (range) of gene mRNA expression levels GCs as compared to non tumoral gastric tissue (PT normalized to 1); *p* value ^a^ (Mann Whitney’s U test); Significant *p*-value ^a^ < 0.05 (in bold); median range of genes between GC subtypes, *p* value (Mann Withney pliciter le test). Significant *p*-value < 0.05 (in bold). NS, not significant. Comparative basal levels of genes in normal tissue (x1) are as follow: *PD1* (19), *PDL1* (53), *PDL2* (86), *IDO1* (70), *TDO2* (18) and *IDO2* (1).

Genes	PT (*n* = 11)	All Tumors (*n* = 29)	*p*-Value ^a^	Intestinal-GC vs. PT (*n* = 16)	*p*-Value ^a^	Diffuse-GC vs. PT (*n* = 13)	*p*-Value ^a^	*p*-Value subtypes
** *Immunity* **								
**PD/PDCD1**	1 (0.35–2.80)	**1.63 (0.81–5.3)**	**0.001**	**1.71 (0.81–5.30)**	**0.003**	**1.53 (0.87–3.08)**	**0.009**	0.35 (NS)
**PDL1/CD274**	1 (0.57–2.70)	1.27 (0.52–6.63)	0.33 (NS)	1.46 (0.52–6.63)	0.08 (NS)	1.03 (0.57–1.56)	0.84 (NS)	**0.018**
**PDL2/PDCDL2**	1 (0.54–1.63)	**1.52 (0.7–2.84)**	**0.009**	1.67 (0.70–2.79)	**0.014**	1.21 (0.89–2.84)	**0.036**	0.51 (NS)
** *Trypt metabolism* **								
**IDO1**	1 (0.19–1.46)	**2.17 (0.34–205)**	**<0.0001**	**3 (0.34–205)**	**0.0006**	1.96 (0.57–4.78)	**0.002**	0.14 (NS)
**TDO2**	1 (0.45–2.95)	**5.41 (1.36–25.2)**	**<0.0001**	**7.45 (1.4–25.2)**	**<0.0001**	**3.33 (1.36–11.9)**	**0.0002**	**0.049**
** *Arylhydrocarbon receptor* **								
**AhR**	1 (0.37–1.64)	**1.94 (0.55–3.53)**	**0.002**	**1.60 (0.65–3.53)**	**0.003**	**2.12 (0.55–3.35)**	**0.001**	0.13 (NS)

**Table 3 biomedicines-10-00240-t003:** Correlation of genes involved in immune checkpoints (*PD1, PD-L1* and *PD-L2*) with clinical parameters in all gastric tumors and subtypes. Median (range) of gene mRNA expression levels; *p* value (Mann Whitney). * Significant *p* value < 0.05 (in bold). ND, not determined, EPN, perineural invasion, TNM, tumor, node, metastasis.

All Gastric Tumors (*n* = 29)				Intestinal Sub-Type (*n* = 16)				Diffuse Sub-Type (*n* = 13)			
	*PD1*	*PDL1*	*PDL2*		*PD1*	*PDL1*	*PDL2*		*PD1*	*PDL1*	*PDL2*
**Gender.**	*p* = 0.25	*p* = 0.51	*p* = 0.65	**Gender.**	*p* = 0.11	*p* = 0.58	*p* = 0.15	**Gender.**	*p* = 0.81	*p* = 0.80	*p* = 0.19
**Male (*n* = 13)**	1.5 (0.8–3.1)	1.14 (0.5–6.6)	1.21 (0.7–2.8)	**Male (*n* = 7)**	1.5 (0.81–3.13)	1.4 (0.52–6.63)	1.17 (0.7–2.65)	**Male (*n* = 6)**	1.51 (0.87–3.08)	1.04 (0.62–1.27)	1.71 (0.89–2.84)
**Female (*n* = 16)**	1.7 (1.2–5.3)	1.33 (0.6–5.5)	1.53 (0.9–2.8)	**Female (*n* = 9)**	2.18 (1.42–5.3)	2.03 (0.78–5.55)	1.83 (1.12–2.8)	**Female (*n* = 7)**	1.58 (1.22–2.01)	1.03 (0.57–1.56)	1.16 (0.89–1.83)
**Age**	*p* = 0.08	*p* = 0.18	*p* = 0.20	**Age**	ND	ND	ND	**Age**	*p* = 0.72	*p* = 0.12	*p* = 0.80
**<60 years (*n* = 9)**	1.49 (0.8–2.0)	1.09 (0.5–1.6)	1.21 (0.8–1.8)	**<60 years (*n* = 1)**	0.81	0.52	0.76	**<60 years (*n* = 8)**	1.51 (0.87–2.01)	1.11 (0.62–1.56)	1.38 (0.89–1.83)
**>60 years (*n* = 20)**	1.71 (0.9–5.3)	1.33 (0.6–6.6)	1.67 (0.7–2.8)	**>60 years (*n* = 15)**	1.76 (0.94–5.3)	1.52 (0.8–6.63)	1.83 (0.7–2.8)	**>60 years (*n* = 5)**	1.58 (1.22–3.08)	0.78 (0.57–1.27)	1.16 (0.90–2.84)
**Tumor invasion**	*p* = 0.74	*p* = 0.72	*p* = 0.32	**Tumor invasion**	*p* = 0.86	*p* > 0.9999	*p* > 0.9999	**Tumor invasion**	ND	ND	ND
**T1-T2 (*n* = 6)**	1.42 (1–5.3)	1.14 (0.6–85)	0.93 (0.7–2.8)	**T1-T2 (*n* = 4)**	2.85 (0.94–5.3)	1.67 (0.96–4.25)	1.66 (0.7–2.8)	**T1-T2 (*n* = 2)**	1.42 (1.27–1.58)	0.67 (0.57–0.78)	0.93 (0.9–0.96)
**T3-T4 (*n* = 23)**	1.65 (0.8–3.1)	1.27 (0.5–6.6)	1.55 (0.8–2.8)	**T3-T4 (*n* = 12)**	1.71 (0.81–3.13)	1.46 (0.52–6.63)	1.67 (0.76–2.65)	**T3-T4 (*n* = 11)**	1.53 (0.9–3.1)	1.1 (0.62–1.56)	1.55 (0.89–2.84)
**Vascular invasion**	*p* = 0.23	*p* = 0.48	***p* = 0.04 ***	**Vascular invasion**	*p* = 0.14	*p* = 0.56	***p* = 0.03 ***	**Vascular invasion**	*p* > 0.9999	*p* = 0.83	*p* = 0.79
**negative (*n* = 9)**	1.63 (1.3–5.3)	1.56 (0.6–6.6)	**1.83 (1.2–2.8)**	**negative (*n* = 6)**	2.66 (1.42–5.3)	3.07 (0.78–6.63)	**2.16 (1.21–2.8)**	**negative (*n* = 3)**	1.53 (1.32–1.63)	0.88 (0.62–1.56)	1.21 (1.18–1.72)
**positive (*n* = 20)**	1.61 (3.1–5.3)	1.22 (0.5–5.5)	**1.17 (0.7–2.8)**	**positive (*n* = 10)**	1.65 (0.81–2.5)	1.37 (0.52–5.55)	**1.17 (0.7–2.65)**	**positive (*n* = 10)**	1.53 (0.87–3.08)	1.06 (0.57–1.35)	1.35 (0.89–2.84)
**Lymphatic invasion**	***p* = 0.05**	***p* = 0.009 ***	*p* = 0.72	**Lymphatic invasion**	*p* = 0.054	***p* = 0.04 ***	*p* = 0.39	**Lymphatic invasion**	ND	ND	ND
**negative (*n* = 11)**	**2.18 (0.9–3.1)**	**2.03 (0.6–6.6)**	1.21 (0.7–2.8)	**negative (*n* = 10)**	2.24 (0.94–5.3)	**3.07 (0.96–6.63)**	1.36 (0.7–2.8)	**negative (*n* = 1)**	1.53	0.62	1.18
**positive (*n* = 17)**	**1.49 (0.8–3.1)**	**1.03 (0.5–1.6)**	1.71 (0.8–2.9)	**positive (*n* = 5)**	1.48 (0.81–1.66)	**0.87 (0.52–1.52)**	2.22 (0.76–2.65)	**positive (*n* = 12)**	1.53 (0.87–3.08)	1.06 (0.57–1.56)	1.38 (0.89–2.84)
**Metastasis**	*p* = 0.25	*p* = 0.15	*p* = 0.20	**Metastasis**	ND	ND	ND	**Metastasis**	*p* = 0.93	*p* = 0.82	*p* = 0.79
**negative (*n* = 24)**	1.65 (0.9–5.3)	1.29 (0.6–6.6)	1.71 (0.7–2.8)	**negative (*n* = 15)**	1.76 (0.94–5.3)	1.52 (0.8–6.63)	1.83 (0.7–2.8)	**negative (*n* = 9)**	1.49 (0.87–3.08)	1.03 (0.57–1.32)	1.71 (0.89–2.84)
**positive (*n* = 5)**	1.53 (0.8–2)	0.72 (0.5–1.6)	1.18 (0.8–1.5)	**positive (*n* = 1)**	0.81	0.52	0.76	**positive (*n* = 4)**	1.58 (1.22–2.01)	1.03 (0.62–1.56)	1.19 (1.16–1.55)
**TNM**	*p* = 0.37	***p* = 0.008 ***	*p* = 0.43	**TNM**	***p* = 0.037 ***	***p* = 0.03 ***	*p* = 0.33	**TNM**	*p* = 0.21	*p* = 0.52	*p* = 0.80
**I-II (*n* = 16)**	1.8 (0.9–5.3)	**1.33 (0.9–6.6)**	1.36 (0.7–2.8)	**I-II (*n* = 11)**	**2.18 (0.94–5.3)**	**2.03 (0.96–6.63)**	1.21 (0.7–2.8)	**I-II (*n* = 5)**	1.32 (0.87–1.9)	1.09 (0.88–1.32)	1.71 (0.89–1.83)
**III-IV (*n* = 13)**	1.53 (0.8–3.1)	**0.87 (0.5–1.6)**	1.55 (0.76–2.8)	**III-IV (*n* = 5)**	**1.48 (0.81–1.66)**	**0.87 (0.52–1.52)**	2.22 (0.76–2.65)	**III-IV (*n* = 8)**	1.6 (1.22–3.08)	0.88 (0.57–1.56)	1.19 (0.9–2.84)
**EPN**	*p* = 0.13	*p* > 0.999	*p* = 0.54	**EPN**	*p* = 0.02 *	*p* = 0.86	*p* = 0.11	**EPN**	ND	ND	ND
**negative (*n* = 6)**	2.1 (1.3–5.3)	1.26 (0.6–4.25)	1.67 (0.9–2.8)	**negative (*n* = 4)**	3.5 (1.76–5.3)	1.69 (1.18–4.25)	2.16 (1.52–2.8)	**negative (*n* = 2)**	1.42 (1.27–1.58)	0.67 (0.57–0.78)	0.93 (0.9–0.96)
**positive (*n* = 23)**	1.53 (0.8–3.1)	1.29 (0.5–6.6)	1.21 (0.7–2.8)	**positive (n= 12)**	1.57 (0.81–3.13)	1.46 (0.52–6.63)	1.19 (0.7–2.65)	**positive (*n* = 11)**	1.53 (0.9–3.1)	1.1 (0.62–1.56)	1.55 (0.89–2.84)
**Smoking**	*p* = 0.75	*p* = 0.91	*p* = 0.0006 *	**Smoking**	*p* = 0.45	*p* = 0.46	*p* = 0.0003 *	**Smoking**	*p* = 0.63	*p* = 0.57	*p* = 0.74
**negative (*n* = 12)**	1.58 (0.9–5.3)	1.26 (0.6–4.2)	1.83 (0.9–2.6)	**negative (*n* = 8)**	1.71 (1.42–5.3)	1.37 (0.78–4.25)	2.05 (1.52–2.65)	**negative (*n* = 4)**	1.51 (0.87–1.63)	1.11 (0.62–1.56)	1.19 (0.89–1.71)
**positive (*n* = 10)**	1.61 (0.8–3.1)	1.33 (0.5–6.6)	1.04 (0.7–1.5)	**positive (*n* = 7)**	1.65 (0.81–3.13)	1.54 (0.52–6.63)	1.12 (0.7–1.21)	**positive (*n* = 3)**	1.58 (1.27–2.01)	0.78 (0.57–1.35)	0.96 (0.9–1.55)

**Table 4 biomedicines-10-00240-t004:** Correlation of genes involved in tryptophan metabolism with clinical parameters in all gastric tumors and subtypes. Median (range) of gene mRNA expression levels; *p* value (Mann Whitney). * Significant *p* value <0.05 (in bold). ND, not determined.

All Gastric Tumors	IDO1	TDO2	Intestinal Sub-Type (*n* = 16)	IDO1	TDO2	Diffuse Sub-Type (*n* = 13)	IDO1	TDO2
**Gender.**	*p* = 0.23	*p* = 0.08	**Gender.**	*p* = 0.09	*p* = 0.53	**Gender.**	*p* = 0.71	***p* = 0.035 ***
**Male (*n* = 13)**	2 (0.34–44.5)	8.5 (1.4–25)	**Male (*n* = 7)**	1.77 (0.34–44.5)	9.45 (1.38–25)	**Male (*n* = 6)**	2.37(0.57–4.78)	**6.82 (1.92–1.9)**
**Female (*n* = 16)**	2.25 (1.3–205)	4.23 (1.4–20)	**Female (*n* = 9)**	6.36 (1.43–205)	6.19 (2.33–20)	**Female (*n* = 7)**	1.75(1.31–3.43)	**2.42 (1.36–5.26)**
**Age**	*p* = 0.06	***p* = 0.045 ***	**Age**	ND	ND	**Age**	*p* = 0.72	*p* = 0.78
**<60 years (*n* = 9)**	1.96 (0.3–3.4)	**3.33 (1.4–7.2)**	**<60 years (*n* = 1)**	0.34	1.38	**<60 years (*n* = 8)**	1.98 (0.57–0.43)	4.16 (1.92–7.17)
**>60 years (*n* = 20)**	2.85 (1–205)	**7.45 (1.4–25.2)**	**>60 years (*n* = 15)**	3.15 (0.98–205)	8.5 (2.33–25.2)	**>60 years (*n* = 5)**	1.75 (1.32–4.78)	2.42(1.36–11.9)
**Tumor invasion**	*p* = 0.38	*p* = 0.74	**Tumor invasion**	*p* = 0.32	*p* > 0.9999	**Tumor invasion**	ND	ND
**T1–T2 (*n* = 6)**	3.53 (1.3–53)	5.1 (1.6–20)	**T1–T2 (*n* = 4)**	5.14 (3.15–53)	6.95 (4.73–20)	**T1–T2 (*n* = 2)**	1.4 (1.32–1.5)	2 (1.6–2.4)
**T3–T4 (*n* = 23)**	2 (0.3–205)	6.19 (1.4–25.2)	**T3–T4 (*n* = 12)**	2.05 (0.34–205)	7.7 (1.38–25)	**T3–T4 (*n* = 11)**	2 (0.57–4.8)	5 (1.36–11.9)
**Vascular invasion**	*p* = 0.33	*p* = 0.39	**Vascular invasion**	*p* = 0.18	*p* = 0.22	**Vascular invasion**	*p* = 0.32	*p* = 0.50
**negative (*n* = 9)**	3.43 (0.6–205)	6.19 (1.9–20)	**negative (*n* = 6)**	25.4 (1.43–205)	9.22 (5.4–20)	**negative (*n* = 3)**	0.81 (0.57–3.43)	2.91 (1.92–5.0)
**positive (*n* = 20)**	1.98 (0.3–141)	5.19 (1.4–25.2)	**positive (*n* = 10)**	2.05 (0.34–141)	5.78 (1.38–25)	**positive (*n* = 10)**	1.98(1.31–4.78)	4.29 (1.36–11.9)
**Lymphatic invasion**	***p* = 0.005 ***	*p* = 0.85	**Lymphatic invasion**	***p* = 0.004 ***	*p* = 0.37	**Lymphatic invasion**	ND	ND
**negative (*n* = 11)**	**6.3 (0.57–205)**	5.4 (1.9–20)	**negative (*n* = 10)**	**25.4 (1.74–205)**	5.91 (2.33–20)	**negative (*n* = 1)**	0.57	1.92
**positive (*n* = 17)**	**1.77 (0.3–4.8)**	5.2 (1.4–25.2)	**positive (*n* = 5)**	**1.43 (0.34–2.96)**	12.47 (1.38–25)	**positive (*n* = 12)**	1.98 (0.8–4.8)	4.16 (1.36–11.9)
**Metastasis**	*p* = 0.12	***p* = 0.005 ***	**Metastasis**	ND	ND	**Metastasis**	*p* = 0.75	***p* = 0.034 ***
**negative (*n* = 24)**	2.53 (0.8–205)	**6.44 (1.6–25)**	**negative (*n* = 15)**	3.15 (0.98–205)	8.5 (2.33–25)	**negative (*n* = 9)**	2 (0.81–4.78)	**5.26 (1.58–11.9)**
**positive (*n* = 5)**	1.75 (0.3–3.4)	**1.92 (1.4–2.9)**	**positive (*n* = 1)**	0.34	1.38	**positive (*n* = 4)**	0.85 (0.57–3.43)	**2 (1.36–2.91)**
**TNM**	***p* = 0.035 ***	*p* = 0.44	**TNM**	***p* = 0.02 ***	*p* = 0.38	**TNM**	*p* = 0.78	*p* = 0.17
**I–II (*n* = 16)**	**3.25 (0.8–205)**	5.91 (2.3–20)	**I–II (*n* = 11)**	**6.36 (0.98–205)**	6.41 (2.33–20)	**I–II (*n* = 5)**	2 (0.81–3.35)	5.26 (3.33–7.17)
**III–IV (*n* = 13)**	**1.75 (0.3–4.8)**	2.91 (1.4–25.2)	**III–IV (*n* = 5)**	**1.43 (0.34–2.96)**	12.47 (1.38–25)	**III–IV (*n* = 8)**	1.85(0.57–4.78)	2.24(1.36–11.9)
**EPN**	*p* = 0.94	*p* = 0.22	**EPN**	*p* = 0.86	*p* = 0.34	**EPN**	ND	ND
**negative (n= 6)**	2.03 (1.3–53)	2.49 (1.6–20)	**negative (n= 4)**	4.34 (1.74–53)	3.99 (2.33–20)	**negative (*n* = 2)**	1.4 (1.32–1.5)	2 (1.6–2.4)
**positive (n= 23)**	2.17 (0.3–205)	6.41 (1.4–25.2)	**positive (n= 12)**	3.05 (0.34–205)	8.75 (1.38–25)	**positive (*n* = 11)**	2 (0.57–4.8)	5 (1.36–11.9)
**Smoking**	*p* = 0.97	*p* = 0.18	**Smoking**	*p* = 0.68	*p* = 0.28	**Smoking**	*p* = 0.40	*p* = 0.23
**negative (*n* = 12)**	2.16 (0.6–205)	6.82 (1.9–25)	**negative (*n* = 8)**	2.05 (1.3–205)	10.74 (2.33–25)	**negative (*n* = 4)**	2.67(0.57–3.43)	4.69(1.92–7.17)
**positive (*n* = 10)**	2.55 (0.3–141)	4.93 (1.4–10)	**positive (*n* = 7)**	3.92 (0.34–141)	6.41 (1.38–10)	**positive (*n* = 3)**	1.48(1.32–1.96)	2.07(1.58–2.42)

**Table 5 biomedicines-10-00240-t005:** Relationship between AhR transcript levels and classical clinical parameters in all GCs and subtypes. Median (range) of gene mRNA expression levels; *p* value (Mann Whitney). ND, not determined.

All Tumors *n* = 29		Intestinal Sub-Type *n* = 16		Diffuse Sub- Type *n* = 13	
	*AhR*		*AhR*		*AhR*
**Gender.**	*p* = 0.51	**Gender.**	*p* = 0.19	**Gender.**	*p* = 0.81
**Male (*n* = 13)**	1.54 (0.55–3.33)	**Male (*n* = 7)**	1.71 (0.89–2.84)	**Male (*n* = 6)**	1.51 (0.87–3.08)
**Female (*n* = 16)**	1.35 (0.65–3.53)	**Female (*n* = 9)**	1.16 (0.89–1.83)	**Female (*n* = 7)**	1.58 (1.22–2.01)
**Age**	*p* = 0.82	**Age**	ND	**Age**	*p* = 0.72
**<60 years (*n* = 9)**	1.94 (0.55–3.35)	**<60 years (*n* = 1)**	1.38 (0.89–1.83)	**<60 years (*n* = 8)**	1.51 (0.87–2.01)
**>60 years (*n* = 20)**	1.80 (0.65–3.53)	**>60 years (*n* = 15)**	1.16 (0.90–2.84)	**>60 years (*n* = 5)**	1.58 (1.22–3.08)
**Tumor invasion**	*p* = 0.21	**Tumor invasion**	ND	**Tumor invasion**	ND
**T1–T2 (*n* = 6)**	1.45 (0.65–2.86)	**T1–T2 (*n* = 4)**	0.93 (0.9–0.96)	**T1–T2 (*n* = 2)**	1.42 (1.27–1.58)
**T3–T4 (*n* = 23)**	1.94 (0.55–3.53)	**T3–T4 (*n* = 12)**	1.55 (0.89–2.84)	**T3–T4 (*n* = 11)**	1.53 (0.9–3.1)
**Vascular invasion**	*p* = 0.14	**Vascular invasion**	*p* = 0.79	**Vascular invasion**	*p*>0.9999
**negative (*n* = 9)**	1.25 (0.82–2.96)	**negative (*n* = 6)**	1.21 (1.18–1.72)	**negative (*n* = 3)**	1.53 (1.32–1.63)
**positive (*n* = 20)**	2.05 (0.55–3.53)	**positive (*n* = 10)**	1.35 (0.89–2.84)	**positive (*n* = 10)**	1.53 (0.87–3.08)
**Lymphatic invasion**	*p* = 0.11	**Lymphatic invasion**	ND	**Lymphatic invasion**	ND
**negative (*n* = 11)**	1.25 (0.65–3.18)	**negative (*n* = 10)**	1.18	**negative (*n* = 1)**	1.53
**positive (*n* = 17)**	2.11 (0.55–3.53)	**positive (*n* = 5)**	1.38 (0.89–2.84)	**positive (*n* = 12)**	1.53 (0.87–3.08)
**Metastasis**	*p* = 0.92	**Metastasis**	ND	**Metastasis**	*p* = 0.93
**negative (*n* = 24)**	1.96 (0.55–3.53)	**negative (*n* = 15)**	1.71 (0.89–2.84)	**negative (*n* = 9)**	1.49 (0.87–3.08)
**positive (*n* = 5)**	2.1 (0.82–2.55)	**positive (*n* = 1)**	1.19 (1.16–1.55)	**positive (*n* = 4)**	1.58 (1.22–2.01)
**TNM**	*p* = 0.30	**TNM**	*p* = 0.80	**TNM**	*p* = 0.21
**I–II (*n* = 16)**	1.77 (0.54–3.35)	**I–II (*n* = 11)**	1.71 (0.89–1.83)	**I–II (*n* = 5)**	1.32 (0.87–1.9)
**III–IV (*n* = 13)**	2.11 (0.82–3.53)	**III–IV (*n* = 5)**	1.19 (0.9–2.84)	**III–IV (*n* = 8)**	1.6 (1.22–3.08)
**EPN**	*p* = 0.47	**EPN**	ND	**EPN**	ND
**negative (*n* = 6)**	1.58 (0.90–2.86)	**negative (*n* = 4)**	0.93 (0.9–0.96)	**negative (*n* = 2)**	1.42 (1.27–1.58)
**positive (*n* = 23)**	1.94 (0.55–3.53)	**positive (*n* = 12)**	1.55 (0.89–2.84)	**positive (*n* = 11)**	1.53 (0.9–3.1)
**Smoking**	*p* = 0.85	**Smoking**	*p* = 0.74	**Smoking**	*p* = 0.63
**negative (*n* = 12)**	1.77 (0.55–3.35)	**negative (*n* = 8)**	1.19 (0.89–1.71)	**negative (*n* = 4)**	1.51 (0.87–1.63)
**positive (*n* = 10)**	1.85 (0.82–3.53)	**positive (*n* = 7)**	0.96 (0.9–1.55)	**positive (*n* = 3)**	1.58 (1.27–2.01)

## Data Availability

Data are all contained within the article.

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
