# Peer review of "Differential Expression of Genes Involved in Metabolism and Immune Response in Diffuse and Intestinal Gastric Cancers, a Pilot Ptudy"

_biomedicines, 2022, doi:10.3390/biomedicines10020240_

Round 1

Reviewer 1 Report

I read with interest the work of Perrot-Applanat et al. They provided a pilot study by studying the expression levels of 7 genes involved in metabolism and immune response of diffuse and intestinal gastric cancers.

Due do their low number of patients, the authors propose this study as a pilot study, both in their objective and in the discussion section, thus I find it mandatory, to include in the paper’s title also.

The abstract in not very well structured and should include the p values, where they were found statistically significant.

The study limitations should be introduced more thoroughly, just before the conclusion paragraph.

Author Response

Dear Referee,

Thanks you for your comments and suggestions, each of which is addressed with our responses.

*Description of the methods can be improved.

  Details have been added to the methods description.

*Due do their low number of patients, the authors propose this study as a pilot study, both in their objective and in the discussion section, thus I find it mandatory, to include in the paper’s title also.

  The title now includes ‘a pilot study’.

*The abstract in not very well structured and should include the p values, where they were found statistically significant.

  The abstract has been rewritten to take into account the Referee’s comment.

*Moderate English changes required.

  Minor revisions suggested by a native English-speaking colleague (Peter Brooks) have been adopted,   and his name has now been added in the Acknowledgments.

Reviewer 2 Report

Dear Editor, thank you so much for inviting me to revise this manuscript about gastric cancer.

This study addresses a current topic.

The manuscript is quite well written and organized. English could be improved.

Figures and tables are comprehensive and clear.

The introduction explains in a clear and coherent manner the background of this study.

We suggest the following modifications:

  • Introduction section: although the authors correctly included important papers in this setting, we believe a couple of studies should be cited within the introduction (PMID: 31793342; PMID: 31315643; PMID: 33916206), only for a matter of consistency. We think it might be useful to introduce the topic of this interesting study.
  • Methods and Statistical Analysis: nothing to add.
  • Discussion section: Very interesting and timely discussion. Of note, the authors should expand the Discussion section, including a more personal perspective to reflect on. For example, they could answer the following questions – in order to facilitate the understanding of this complex topic to readers: what potential does this study hold? What are the knowledge gaps and how do researchers tackle them? How do you see this area unfolding in the next 5 years? We think it would be extremely interesting for the readers.

However, we think the authors should be acknowledged for their work. In fact, they correctly addressed an important topic in gastric cancer, the methods sound good and their discussion is well balanced.

One additional little flaw: the authors could better explain the limitations of their work, in the last part of the Discussion.

We believe this article is suitable for publication in the journal although some revisions are needed. The main strengths of this paper are that it addresses an interesting and very timely question and provides a clear answer, with some limitations.

We suggest a linguistic revision and the addition of some references for a matter of consistency. Moreover, the authors should better clarify some points.

Author Response

Dear Referee,

Thanks you for your comments and suggestions, each of which is addressed with our responses.  

*Introduction section: although the authors correctly included important papers in this setting, we believe a couple of studies should be cited within the introduction (PMID: 31793342; PMID: 31315643; PMID: 33916206), only for a matter of consistency. We think it might be useful to introduce the topic of this interesting study.

Following the reviewer’s recommendations, we have now cited two of these papers (Ricci et al 2021 and Xiang et al 2019) in the Introduction, and the third is now cited in the Discussion (Rizzo et al 2019). 

*Very interesting and timely discussion. Of note, the authors should expand the Discussion section, including a more personal perspective to reflect on. For example, they could answer the following questions – in order to facilitate the understanding of this complex topic to readers:

- What potential does this study hold?

We provided a pilot study by studying the expression levels of 7 genes involved in metabolism (IDO1 and TDO2) and immune response of diffuse and intestinal gastric cancers. These genes may be partly regulated through different AhR signaling.While the present study suggests an inactive immune status in the advanced diffuse GC, a better understanding of the complexity of these regulations through different AhR ligands using in vivo and in vitro studies will help to develop agonist/antagonists on animal models. In future, intra-operative peritoneal lavage samples (Tanaka 2021) may also help in order to identify metabolic, extracellular matrix/fibrosis and other signatures involved in peritoneal metastasis. We have now introduced this comment in the Discussion.

-What are the knowledge gaps and how do researchers tackle them?

We have not discriminated the subpopulation of diffusely infiltrating type of GC associated with extensive fibrosis (linitis) (Henson 2004, Jezequel 2010) as compared to non linitis diffuse GC. Further studies in a larger series of gastric tumor samples, especially with different clinical characteristics (early diffuse subpopulation) and linitis (SRCC, known as an increased risk of developing peritoneal metastasis, Honore 2013) would offer opportunity to confirm genes of interest in agressive GC.

These comments have been introduced in the paragraph before the conclusion (Discussion)

-How do you see this area unfolding in the next 5 years? We think it would be extremely interesting for the readers.

The role of the surrounding tumor environment in gastric cancer is particularly important in tumor progression and metastasis. Actually, the microenvironment is a limitation factor for any drug penetration. The more we will be able to understand the different mechanisms implicated, the more we will be able to develop new therapeutic solutions. We can hope that in 5 years the classification of gastric cancer will be changed and the clinician could offer a specific microenvironment dedicated drug for the patient. We now include longer term perspectives in the final conclusion.

*We think the authors should be acknowledged for their work.

 We thanks the reviewer for this comment.

*One additional little flaw: the authors could better explain the limitations of their work, in the last part of the Discussion.

We agree that our study has some limitations. The small number of tumor samples (30) could be a limiting factor and could induce a bias between intestinal and diffuse subtypes. However, in a previous study on the same western cohort (Perrot-Applanat et al. 2019), we observed comparable results with decrease or increase gene expression (such as CDH1, CXCR4 and TGFb involved in epithelial mesenchymal transition and chemotaxis), as now well described. In addition, the number of tumor samples should include –early and –agressive SRCC diffuse GCs.

Limitations of the study have been introduced in the last part of the Discussion, just before the conclusion.  

In addition, the manuscript has been checked by a native English-speaking colleague (Peter Brooks) and his name has now been added in the Acknowledgments.

Round 2

Reviewer 1 Report

The authors made the necessary changes, and I endorse the paper for publication.

Reviewer 2 Report

The authors modified the manuscript according to our suggestions.

We recommend Acceptance.